# The Construction and Validation of a Sustainable Tourism Development Evaluation Model

**DOI:** 10.3390/ijerph17197306

**Published:** 2020-10-07

**Authors:** Han-Shen Chen

**Affiliations:** 1Department of Health Diet and Industry Management, Chung Shan Medical University, No. 110, Sec. 1, Jianguo N. Rd., Taichung City 40201, Taiwan; allen975@csmu.edu.tw; Tel.: +886-4-2473-0022 (ext. 12225); Fax: +886-4-2324-8188; 2Department of Medical Management, Chung Shan Medical University Hospital, No. 110, Sec. 1, Jianguo N. Rd., Taichung City 40201, Taiwan

**Keywords:** climate change, environmental impact, environmental planning, willingness to pay, virtual reality

## Abstract

As climate change, food crises, sustainable development, and ecological conservation gain traction, the revival of traditional fishing villages has become an important governmental policy for Taiwan. To reduce cognitive bias, the choice experiment method was applied to construct an attribute function in fishing village tourism coupled with virtual reality headsets. Conditional logit and random parameter logit models were employed to estimate tourism utility functions. Moreover, a latent class model was employed to determine whether hetxerogeneous preferences regarding fishing village travel existed. The sampling sites were distributed across the Dongshi area. In total, 612 tourists and 170 local residents were interviewed. After incomplete questionnaires were removed, 816 valid questionnaires remained, representing 95.83% of the total questionnaires. Older residents and residents with shorter histories of education were inclined to increase land development and utilization by reducing natural landscapes; tourists preferred preserving landscapes and preventing land development. Residents with more education believed that local landscape imagery was essential. Tourists who were more educated, with high incomes, and those who were older believed that a selling platform incorporating local industries and products within the villages would be attractive for other tourists.

## 1. Introduction

Taiwan is a typical island country with 224 fishing ports. Fishing villages clustered around the ports as corresponding industries have flourished. However, due to depleting fishery resources, the pollution of the marine ecological environment, and the enforcement of the United Nations Convention on the Law of the Sea, the fishery economy witnessed a decline. The prosperity of these villages has also contracted because of the dearth of employment opportunities, thus further engendering the emigration of young residents and leaving behind a largely aging population in the fishing communities. The rural and remote nature of these villages, coupled with dated public facilities and constructions, only further exacerbates the problem. To increase their value, agricultural and fishing villages across the globe actively explore solutions for the socioeconomic problems that they encounter. These villages propose these solutions to induce industrial transformation and enhancement, to strike a balance between the development and preservation of resources, and to ensure the sustainable development of agricultural and fishing villages [1,2,3,4]. Keeping in mind the increasing significance of issues such as global climate change, food crises, sustainable development, and ecological conservation, the revival of fishing villages with traditional features becomes an important policy for the government.

The authorities introduced new plans, such as “Building Rich and Beautiful Agricultural and Fishing Villages,” “New Style in Agricultural Villages,” and “Agricultural Villages Revival 2.0,” with the hope of accelerating agricultural and fishery transformation by combining local industries and culture in a bid to solve the existing problems. With special environmental attributes, such as natural landscapes, precious wildlife, and unique settlement cultures [5,6], tourism is often considered to provide a significant impetus for the development of agricultural and fishing villages [7,8]. It has been suggested that as a new form of tourism with multiple features, fishery tourism will become one of the main practices helming the transformation of coastal areas [9]. Fishery tourism can accelerate economic growth and resolve the livelihood problems of local residents as well as, more importantly, maintain a balance between the development of fishery tourism and the preservation of ocean resources while allowing for sustainable development.

The tourism industry contributed 8.8 trillion USD, i.e., nearly 10.40% of the global gross domestic product (GDP), in 2018, and it has created 319 million jobs across the world [10]. Furthermore, according to the Tourism Bureau Survey of Travel by R.O.C. Citizens (2018) [11], from 2001 to 2018, domestic travel by citizens increased by 75.58%, from 97.45 million to 171.09 million. Instead of seeking comfort in travel, travel by citizens looking to be close to nature increased from 64.70% to 75.60%. Other tourism destinations may include forest recreation areas and leisure farms. Compared to them, fishery tourism, which contributes to ecotourism by combining local natural resources and cultural celebrations as well as experiences in fishery activities and village folk customs, is not limited to providing leisure and entertainment to the public; rather, fishery tourism also provides economic benefits for villagers. Therefore, this serves as a critical juncture for fishing village development.

Overseas fishing villages proactively promote transformation and upgrade at the same time. Misali Island in the United Republic of Tanzania established a marine protected area and developed ecotourism to preserve ocean resources and engage in sustainable development [12]. Further, Andros Island of the Bahamas, housing one-third of the world’s precious wildlife resources, vigorously promoted ecotourism, and its tourism income indicated an annual growth rate of 78% [13]. Similarly, Sebuyau, a fishing village in Malaysia that relied on agriculture, fishery, and handicraft during earlier times, promoted travel experiences to accelerate the local economy, resulting in a tourism income as high as 65 billion Ringgit (17 billion USD) in 2012 [14]. Padín et al. noted that when Galicia, a community in Spain famed for its seafood throughout Europe, encountered economic hardships due to fishing quotas, it transformed its primary economic source to industries that included exquisite handicraft, small-scale fishing, and tourism [15]. Tourists can fish on boats, spend nights in the villages, and visit fishing museums as a means of experiencing the production of the village, village life, and eco-industry. The authorities also collaborate with the tourism industry to promote water activities, such as glass-bottom boat tours and snorkeling, and thus, this has sharpened the industry’s competitive edge through ecotourism. Additionally, Palau generated 40% of its jobs and 75% of its GDP from ecotourism [16]. Furthermore, Japan recently developed a Satoumi policy to promote active maintenance efforts among coastal fishing villagers and increase socioeconomic value across the coastal community. Nagasaki Prefecture, for example, used its ocean resources and added a novel value to its fishing villages by selling unique heart-shaped oysters to achieve sustainable economic and environmental development [17].

With reference to the aforementioned literature, we understand that whether local or overseas, fishing villages encounter various difficulties and changes in relation to the environment. To ensure continued developments, villages use their unique features to engender transformations and upgrades, promote leisure traveling, and design numerous fishery-related activities, thereby adding value to the villages. Jiménez de Madariaga and García del Hoyo suggested that the development of tourism in fishing villages could redefine fishery culture and usher in multiple new approaches into the economy [18]; nonetheless, cooperation and planning between the government and local communities remains essential for transformation. Mease et al. echoed the aforementioned aspect by recommending the inclusion of relevant stakeholders and the public in addition to governmental guidance to ensure that policies for the development and management of such villages are both effective and continuous [17,19].

Dongshi fishing village, the target research area, suffered from problems such as emigration, aging population, labor shortages, decline in fishery economy and productivity, and long-term ecological damage. Owing to guidance from and planning of the government, the Dongshi fishing village has gradually revived and transformed itself in recent years. By selling valuable fishery products and holding related cultural celebrations, Dongshi has earned a reputation in supplying fish specialties and fetches economic revenue for its surroundings. Experiences on the sea (e.g., oyster farm visiting above water), land–water interface zones (e.g., bird watching), special cultural activities on land (e.g., the fishery cultural season), and experiences related to ecology (e.g., oyster picking) has together made the village a new tourist hotspot. However, this transformation eventually affects the natural environment. Cars and motorcycles produce air and noise pollution, while human activities on the sea damage the habitats of animals and plants. The growing number of tourists negatively affects the local quality of life due to the leftover rubbish, polluted water, and ensuing security problems. Sustainability is threatened because natural resources are prone to being exhausted, thus leaving nothing for sightseeing. Owing to the vulnerability of the ecological environment in agricultural and fishing villages, especially due to limited transport, land, and economy, development in such locations could lead to positive or negative effects on the local environment and economy [20,21]. Mura and Ključnikov suggested that although human activities increase the construction of fishing villages and promote tourism industries and fish farming, the activities also deplete the ecological system’s productivity, worsen the quality of the environment, change the landscape, and endanger ecological systems and biodiversity [22]. There is a crucial lesson in fishery tourism, that is, to maintain village characteristics and views while simultaneously ensuring the benefit of the ecological system under the principle of sustainability. Therefore, the present study took the Dongshi fishing village of Taiwan as an example to discuss the utility of sustainable fishery tourism development.

Sustainable tourism usually develops after precise planning and cautious consideration of revenue and environmental impact in areas housing highly sensitive environments. Consequently, evaluating sustainable tourism through an economic analysis entails managing units with respect to decisions in the planning of the usage or continuous functioning of local ecological resources [16,19]. In the field of recreational efficiency evaluation, several studies have used the travel cost method (TCM) and contingent valuation method (CVM) to assess ecotourism’s utility [23,24,25]. However, because TCM and CVM have some limitations in their application that make it difficult to evaluate products with multiple attributes and standards, the choice experiment method (CEM) has the advantage of being used to explore product preferences with multiple attributes and levels.

CEM can assess use value and non-use value simultaneously as well as define a hypothetical market through questionnaire surveys to investigate public preferences in landscape preservation and natural development, which, in turn, reflects the value of environmental goods or services. As CEM can evaluate multiple attributes and levels, different combinations of alternative plans can be raised according to the important characteristics of non-market goods or services. Respondents can identify appropriate alternative plans in accordance with their preferences and avoid bias in assessment through choice sets in different scenarios [26]. Given these advantages, CEM has been widely applied for the assessment of non-market value in recent years, including studies on wildlife conservation [27,28], wetland restoration [29,30], ecotourism preferences [31,32], coastal areas [33,34,35,36], national parks [37,38], and ecosystem services [39,40,41]. Further, Diafas et al. employed CEM to examine ecosystem changes in rural communities and the efficiency of ecosystem preservation [42]. Ferretti and Gandino used CEM to study the revival of abandoned rural buildings at world heritage sites and estimated the benefits they could generate; the results have been provided to authorities as a reference for exploring new policies [43].

In empirical modeling, conditional logit (CL) models can estimate tourists’ average preferences on multi-attributes of ecotourism and estimate the marginal willingness to pay (MWTP) of ecotourism attribute degree [32,44]. To examine the heterogeneous preferences of respondents and changes in their willingness to pay (WTP), given different attribute levels (attributes include experience in folk customs and culture or ecological activities), a random parameter logit (RPL) model is applied to demonstrate divergence among respondents from diverse backgrounds while facing various specific attribute preferences [45,46,47]. To segment target markets, a latent class model (LCM) is applied to segregate respondents into various groups as well as to investigate and compare the differences between groups, for example, the preferences and attitudes of interviewed tourists toward ecotourism along with their socioeconomic background [48]. From the aforementioned studies, we understand that CL, RPL, LCM, and other empirical models in CEM can be applied for the examination and evaluation of preference in relation to multi-attributes in ecotourism destinations with proven results.

Virtual reality (VR) is a high-technology, simulated system manufactured through computer assistance. A sense of space is created through computing, while substances in reality and digital data are turned into a simulated three-dimensional environment, which can be seen or even touched. Users can choose between various input and output equipment, be submerged into a three-dimensional space, and become direct participants in the virtual scene. Interactive, immersive, and real-time features create a vast difference between VR and traditional 3D animation. VR is now extensively applied in research encompassing virtual tours [49,50], shopping [51,52,53], education [54,55], virtual appreciation [56], visual and audio experiences [57], medical treatment [58,59], and disaster prevention [60,61]. A scene simulated by VR reflects more of the real world and diminishes cognitive bias compared to a description using words [62]. Innocenti suggested that in addition to reducing cognitive bias, contingency variables are strictly under control using VR [63]. In view of this, scholars have started combining applications of VR and CEM. Matthews et al. demonstrated various situations of coastal erosion using VR and employed CEM to exemplify the efficiency of coastal protection by tourist inflow [40]. Rid et al. used VR to illustrate a view of houses and their building styles while applying CEM to estimate consumers’ preferences and weigh future development directions [64].

Extant research examining the economic value of non-market resources primarily focuses on forests, coastal areas, national parks, and naturally protected areas. Few studies examine the value of fishery tourism; no research has ever applied VR to the topic. This study combined CEM and VR as a cross-disciplinary approach to broaden the depth of research on sustainable tourism in fishing villages. The construction of this methodology is based on representativeness and research originality and fills a gap in the research efforts on sustainable fishery tourism development that are lacking. Studies on and contributions to the abovementioned problems will serve as references, thus benefiting researchers who aim to evaluate models for sustainable fishery tourism development and those who practice management and administration policies in fishery tourism.

## 2. Materials and Methods

### 2.1. Description of the Study Area

The Dongshi fishing village is located in western Taiwan, facing the Taiwan Strait. The village has an area of approximately 820,000 square kilometers (as illustrated in Figure 1) and a population of approximately 30,000. Some agricultural lands became unsuitable for farming because of land subsidence and soil salinization, and therefore, they were transformed into fish ponds for fish farming. The 14-km-long coastline is vital for the village’s livelihood, with fish ponds covering around 2140 hectares and offshore fish farming area spanning 1800 hectares. The village contains the largest oyster farm in Taiwan, producing more than one-third of the oysters in Taiwan. Fish farming consumes substantial time and effort but provides limited vacancies and salaries. Consequently, many young adults are driven outside the village for their personal development. The agriculture and fishery industries were further damaged when the Taiwanese market was opened to imports of agricultural and fish products after the country enlisted with the World Trade Organization. In recent years, to solve the economic and social problems in the village, the government collaborated with local organizations to plan and develop fishery tourism proactively and demonstrate the unique style of fishing villages. For example, the largest wetland in Taiwan, Aogu Wetland, which covers 1500 hectares, has become an important habitat for waders and a hotspot for bird watchers. The largest shoal of the country, Waisandingzou, is a major oyster farming area. With considerable promotion in terms of culture, history, travel, and the integration of ecology education, the Dongshi fishing village is now a new tourist destination for ecological experiences, cultural education, and leisure.

### 2.2. Construction of an Utility Model for Fishing Village Travel Preference Attributes

The study applied CEM in the construction of a utility model for fishing village travel preference attributes. The study also employed CL and RPL models to estimate the indirect utility functions of fishery tourism. Using data on socioeconomic backgrounds as well as the cognitive and behavioral perspectives of interviewed tourists, differences in the MWTP of various attributes were examined, while an LCM was employed to determine whether heterogeneous preferences for fishing village travel exist among respondents. Finally, the results from the aforementioned empirical analysis were used to estimate the economic utility of fishery tourism.

CEM is a typical random utility model (RUM) for examining MWTP, given various attributes and levels [65]. Therefore, in a binary model, the utility for the nth respondent was assumed to be the different options (U_ni_) that they meet, and the options would maximize utility as shown in Equation (1):(1)$$U_{\mathrm{ni}}=V_{\mathrm{ni}}+\varepsilon_{\mathrm{ni}}$$
where U_ni_ is the utility of attribute n for person i, V_ni_ is the observable utility function, and ε_ni_ is an error that cannot be observed.

This study expects to examine different preferences and the MWTP of respondents from diverse socioeconomic backgrounds, given various attributes and levels; therefore, an RPL model and an LCM were employed for analysis. The overall utility in the LCM is as follows:(2)$$U_{\mathrm{ni}}=V_{\mathrm{ni}}\left(X_{\mathrm{ni}},S_{n}\right)+\varepsilon_{\mathrm{ni}}$$
where V_ni_ is the utility coefficient of observable variable X_ni_ and respondents’ characteristic S_n_ and represents the respondent’s preference, while ε_ni_ is the error.

The estimation method of the LCM assumes that ƒ(β) distribution is separated, while β is the differentiate value of a finite set. It is assumed that under β, possible values of option *k* are represented by b_1_,…,b_k_*,* probability is represented by S_k_, while β=b_k_. Therefore, the option probability is as follows:(3)$$P_{ni}=\sum_{k-1}^{k}S_{nk}L\left(i|k\right)=S_{nk}L\left(i|k\right)$$

In the above equation, L(i|k) is the CL probability by group, and $S_{nk}$ is the group probability. Under the LCM, an individual can be incorporated into behavioral groups to analyze their preferences and WTP. In this manner, assumptions of homogeneity and heterogeneity within groups can be satisfied, while individual characteristics can be used to calculate group probability. However, to estimate the relative importance of product attributes on value, assuming that the degrees of attribute in alternative plan *j* remain the same, the marginal change in WTP of the *k*th attribute can be given by Equation (4):(4)$$\mathrm{WTP}=\frac{-\beta_{k}}{\beta_{c}}$$

### 2.3. Fishery Tourism Attribute and Level Integration Plan

By referring to related survey reports and studies and interviewing professionals and scholars from different disciplines, this study set “land use planning,” “cultural experience,” “local imagery landscape and architecture,” “product and industry promotion,” and “willingness to pay/leisure attractiveness” as the five attributes. Travel attributes and the view of the fishing village were completely shown using VR to reduce cognitive differences due to words or a single photo, which might have an impact on research results. In an evaluation of efficiency, different measurements are applied to different groups of respondents. The investigation focus for the locals is set as the “willingness to pay,” which concerns maintenance and construction with a foundation’s support. For the tourists, the focus is on “leisure attractiveness,” that is, to weigh their preferences regarding various attributes by assessing how much additional time they would spend on sightseeing, following the inclusion of new features. Table 1 lists the setting and details of these five attributes.

#### 2.3.1. Land Use Planning

Carlsson et al. observed that the maintenance of the natural landscape would significantly and positively affect tourism [66]. Ridding et al. emphasized that keeping and restoring forest land and natural landscapes promote sustainable development in an ecological system [67]. Chen and Chen indicated that tourists and local residents were divided on how to use tourism resources, such as whether to maintain the status quo or develop new land uses [68]. This study took the present conditions as the base level, adding “maintenance of natural landscape” and “land utilization and planning” as two new levels for respondents to choose from.

#### 2.3.2. Fishery Cultural Experience

Demarco suggested that a cultural experience would deepen tourists’ understanding of local culture and history [69]. Chaminuka et al. examined tourist preferences regarding ecotourism development and their MWTP, and the studied experiences included rural accommodation, rural cultural travel, and rural handicraft [70]. While studying cultural travel to temples in Taiwan, Wu et al. discovered that most tourists like interactive experiences, such as understanding the history of temples by watching animations [71]. Increasing fishery cultural experience was thus listed as a level to estimate respondent preferences on cultural experience.

#### 2.3.3. Local Landscape and Architecture Imagery

This explains how producers project compelling site meanings and how tourists decode these meanings through the lens of an imagined community. Oh et al. argue that tourists primarily seek and consume engaging experiences accompanied by the goods and/or service components of the destinations [72]. Thus, by designing and building local landscape and architecture imagery, an understanding of fishing villages via a viewing experience is enhanced. Farrelly used three memorial venues of different imageries to examine the experiences of tourists [73]. In this study, building new local landscape and architecture imagery was listed as a level for respondents to decide on.

#### 2.3.4. Local Products and Industries Promotion

Ferretti and Gandino studied the efficiency of reviving and using abandoned rural buildings at world heritage sites and found that most respondents preferred to turn abandoned buildings into bicycle shops followed by traditional product shops [43]. Grebitus et al. examined the correlation between consumer behavior and agricultural development and concluded that consumers were attracted to fresher and higher-quality local products, which promoted the economy and incentivized industrial development [74]. Kastenholz et al. stated that economic sustainability requires the generation of economic benefits for local communities by providing value to local assets and competences, job creation, and the promotion of local products [75]. Thus, the study included whether to promote local products and industries as a level for respondents.

#### 2.3.5. Willingness to Pay/Leisure Attractiveness

Agimass et al. determined the leisure attractiveness of forests to tourists based on the distance of forests from the respondents [76]. Further, Bakhtiari et al. suggested that the attributes of leisure environment and adding travel services could be examined with distance as a variable, as distance could indicate the degree of attraction that respondents have to a leisure location [77]. However, a minority of respondents had an unclear cognition of distance; consequently, extra travel time was employed instead as an evaluation attribute for tourists to understand the differences in attractiveness caused by changes in tourism factors. Tourist sites help the development of the region and the economy, while residents in the region also have the responsibility to maintain the development of the tourist sites. The study uses the tourism management and maintenance perspective to explore residents’ preferences for the maintenance of tourist attractions through the maintenance fund.

### 2.4. Construction of Evaluation Model for Fishery Tourism Attributes

Orthogonal design is often used as a tool for scaling down experimental programs and can achieve statistical accuracy in the case of scaling down. The study reduced 180 (2^2^ × 3^2^ × 5) combinations to 18 alternative combinations through orthogonal design, and one current solution. After you include the current scenario in each group’s selection set, each scenario contains two scenarios with random alternatives and one current scenario. Each questionnaire contains five selection sets drawn from it, with 34 versions. The combination of the design process and the above selection sets improves the statistical efficiency of the selection set design.

In addition, the model herein was built by panorama photos. Collections of photos taken by a panoramic camera were processed and combined using software to produce panoramic photos in 720 degrees; subsequently, the photos could be turned into VR spaces. The advantages of this building style include speed and a high resemblance to reality, as the user can experience a 720-degree reality view that is all-dimensional. Interactive features are also added to allow for a well-rounded understanding of the destinations’ essence. This can reduce evaluation bias due to information differences in comparison with previous studies that used traditional pictures and descriptions involving words (as illustrated in Figure 2).

### 2.5. Survey Design and Respondents

The questionnaire was divided into three sections: “tourist behavior,” “fishing village attribute preference,” and “basic information of tourists.” The details of the different sections are the following.

(1) Tourist behavior: this section was primarily designed to understand the traveling behavior of tourists, including the frequency of travel, types of activities, travel time, and satisfaction regarding the activities, in order to decipher a basic picture of tourists’ situations and how they graded different conditions of tourism attributes.

(2) Fishing village attribute preference: this section involved applying CEM to examine respondents’ preferences in fishery tourism, including “natural landscape protection,” “fishery cultural experience,” “local imagery landscape and architecture,” “revival of industrial environment,” and “leisure distance.”

(3) Basic information of tourists: this section aimed to identify the socioeconomic background of respondents, including their gender, age, education level, occupation, residence location, and average monthly individual income.

Local residents and tourists were interviewed in early January 2019 using an initial questionnaire. This questionnaire was subsequently modified based on the conservation and management status, expert opinion and advice, and the initial results. Interviews using the final questionnaire were conducted from April to August 2019. Respondents were randomly selected and received an individual on-site interview. Respondents could experience the targeted destinations through VR and fully understood the research attributes in concrete expressions. This prevented information asymmetry confusion due to differences in understanding and cognition. The sampling sites were distributed across the Dongshi area. The respondents were divided into two groups: local residents and tourists. In total, 612 tourists and 170 local residents were interviewed. After incomplete questionnaires were removed, 816 valid questionnaires remained, representing 95.83% of the total questionnaires. (Table 2).

Of all the respondents, 401 respondents (51.30%) were female. Most respondents were in the age group of 31–40 years, with 288 (36.80%) in this group, while 178 respondents (22.80%) were aged 41–50 years. In terms of education, 476 (60.90%) respondents had a bachelor’s degree, and 183 (23.40%) had a master’s degree. Finally, 453 (57.90%) respondents had an average monthly individual income of 25,001–50,000 NTD, while 216 (27.60%) earned under 25,000 NTD.

## 3. Results

### 3.1. Analysis of the Preferences and Benefits of Fishing Village Environmental Resource Attributes

This study adopted CL and RPL models to estimate fishing village travel utility functions (Table 3 and Table 4). The CL model showed that tourists and local residents had noticeable preferences in the designated tourism attributes. To further examine the differences in preferences between the groups of respondents, this study employed an RPL model to analyze the differences in fishing village attribute preferences of tourists and local residents and estimate related essences affecting each attribute. These results can provide implications for the government and concerned parties when planning and implementing environmental policies in the future.

The creation of three kinds of cultural experiences would increase attractiveness to visitors to the highest degree (254 min), followed by natural landscape preservation (184 min), the creation of two kinds of cultural experiences (181 min), product and industry promotion (54 min), and local landscape and architecture imagery (24 min). Overdevelopment of land would decrease the willingness to travel. From the perspective of tourists, the attractiveness of tourism attributes such as cultural experiences, natural landscape preservation, and product and industry promotion was much higher than that of local landscape and architecture imagery, which means that the attractiveness of local landscape and architecture imagery is weak.

Local residents’ WTP was the highest (304 NTD) for the creation of three kinds of cultural experiences, followed by the creation of two kinds of cultural experiences (239 NTD), product and industry promotion (130 NTD), local imagery landscape and architecture (111 NTD), natural landscape preservation (103 NTD), and land development and utilization (72 NTD). From the perspective of local residents, preferences are higher for increasing cultural experiences, product and industry promotion, and creation of local imagery landscape and architecture. Notably, land development and utilization and natural landscape preservation both bore positive results, reflecting a division of opinion in land use within the residents.

### 3.2. Analysis of Fishery Tourism’s Attractive Essence Differentiation

Respondents had different preferences regarding tourism development resources because of their identities and standpoints. Through an examination and comparison via a cross-analysis on their socioeconomic backgrounds, Table 5 and Table 6 illustrate the correlation between respondents’ backgrounds and different resources attributes.

The cross-analysis on tourists’ socioeconomic backgrounds and attribute preferences indicated the following. (1) The two attributes, natural landscape preservation and lowering development and utilization of land, were highly attractive to more educated and younger respondents. (2) Increasing cultural experiences was highly attractive to married respondents under the age of 50 with bachelor’s degrees. (3) Local landscape and architecture imagery was more attractive to older respondents. (4) The incorporation of local products and industries into the fishing village could effectively attract individuals who are more educated and with a higher income over the age of 50.

The cross-analysis on local residents’ socioeconomic backgrounds and attribute preferences showed the following. (1) More educated and younger residents were willing to reduce land development and preserve natural landscapes. (2) Less educated and older residents believed natural landscapes should be reduced while increasing development and utilization of land. (3) More educated residents believed that the local landscape and architecture imagery were essential. (4) Residents from different age groups had divided opinions on incorporating local products and industries into the fishing village.

### 3.3. Analysis on Potential Categories

Through the LCM, this study further analyzed tourism environment preferences of different tourist categories. From Table 7, two potential categories and the difference in tourism environment preferences were recognized. Concerning the attractiveness of tourism environment, the first category of tourists was inclined toward an “increase in cultural experience,” “natural resources preservation,” and “product and industry promotion,” while “imagery architecture” was the least attractive. It is noteworthy that “land development and utilization” created push factors in travel. The second category of tourists was inclined toward “natural resources preservation,” “product and industry promotion,” “imagery architecture,” and “increase in cultural experience,” while their sense of decreasing “land development and utilization” was stronger than that of the first category. The first category comprised 73.10% of the sample. They were clearly inclined toward cultural experiences and product and industry promotion and could be called “tourists for depth.” Contrastingly, the second category expressed higher preferences only for “natural resources preservation,” while expressing indifference to “cultural experiences,” “product and industry promotion,” and “imagery architecture.” This category of tourists, which constituted 26.90% of the sample, can be named “general tourists.” Comparing the socioeconomic backgrounds and travel behaviors of the two categories, “tourists for depth” were mostly married and under the age of 49. In terms of travel behavior, tourists are attracted to more entertainment facilities, more cultural experiences, natural landscape preservation, and the revival of local industries, while an increase in architecture imagery would push them away.

## 4. Discussion

Sustainable tourism is usually regarded as a solution to the environmental impact of tourism, which emphasizes the appeal of environmental conservation [78], and environmentally responsible behavior [79]. Kumar (2002) believes that, in a sensitive and fragile ecosystem, tourism may not come without incurring costs [80]. Although sustainable tourism offers a basis on which many community tourism projects are founded, there is nevertheless an extensive amount of literature documenting the negative impacts of tourism—not only on natural and cultural resources but also on the local community itself, as well as on relations between residents and tourists [81,82]. Tourists and local residents are of paramount importance in tourism development [83], and their preferences and perceptions are of paramount importance among tourism stakeholders. Fishing villages are bounded by limited natural resources, a degree of vulnerability, restricted hazard resilience, and resultant economic dependence. To develop tourism and provide leisure services, Fishing villages must bear the negative effects on the ecological environment and social culture. Thus, the development of tourism resources with careful consideration of the ecology, economy, and social as well as sustainable development while cushioning the impact of leisure is significant.

From the abovementioned results, tourists and residents evidently had different cognitions concerning several tourism environment attributes. Factors for the tourists’ environmental preferences included the following: (1) concerning the development and preservation of natural resources, tourists preferred reducing land development and utilization and supported preserving landscapes; (2) more cultural experiences would attract more visitors; (3) although the creation of local landscape and architecture imagery was still moderately attractive to tourists, their attractiveness was lower than other attributes; and (4) product and industry promotion led to evident positive results, suggesting that when a destination provides more local products, the willingness of tourists to visit increases. The environmental preferences influencing the factors of local residents included the following: (1) concerning the development and preservation of natural resources, some local residents believed in more development and utilization of land, while others supported enhancing landscape preservation; (2) residents agreed to increase cultural experiences to draw in more visitors; (3) residents generally believed that the creation of local landscape and architecture imagery attracts more visitors; and (4) product and industry promotion brought clear positive results, indicating that residents believed that tourist destinations should open more spaces to sell and display local agricultural and fishery products.

The above analysis shows that tourists and local residents have great differences in their environmental preferences. Tourists hope to reduce land development and utilization and supported preserving landscapes, whereas residents hope to reinforce the development and utilization of land. The results of this analysis are same as those of Chen and Chen (2019) [68], which indicated that tourists and local residents were divided on how to use tourism resources. In the future, planners of fishing village development should consider the opinions of tourists and local residents and include their knowledge regarding the environmental resources of travel destinations as well as their needs to construct a more accurate foundation for decision-making [84] such as whether to maintain the status quo or develop new land uses [68]. Stefanica and Butnaru [85] argued that responsibility for the environmental impacts of tourism development should be shared by tourism industry operators and tourists alike. Several studies have reported that tourism creates new opportunities for residents such as new shopping and recreation opportunities [86,87].

Second, some residents believed that deficiencies exist in the current fishing villages tourism environment, which should be overcome by increasing land development and using hardware facilities to expand tourism capacity. They believed that local landscape and architecture imagery can attract visitors but were divided on incorporating local products and industries into the fishing village. However, the factors that further attracted tourists included experiencing the village life, the maintenance of the natural rural landscape, and the incorporation and promotion of local products and industries in the fishing village. Local landscape and architecture imagery was the least attractive attribute to tourists. Previous studies have indicated that a cultural experience would deepen tourists’ understanding of local culture and history [69], and by designing and building local landscape and architecture imagery, an understanding of fishing villages via a viewing experience is enhanced [72]. Furthermore, consider economic sustainability requires the generation of economic benefits for local communities by job creation and the promotion of local products [74,75]. The findings of this study are consistent with the majority of studies especially when it comes to the perceived economic benefits of tourism [88,89]. This is because people firmly believe that tourism is the catalyst for local economic development.

In this study, only five environmental resource attributes are illustrated because of the limitation in expression via panoramic images. However, there are many more environmental resource attributes that can be discussed with regard to fishing villages, such as the maintenance of the leisure environment, car park spaces, the design of shaded walking trails, and guided tours. Through a related evaluation of environment attributes, we can learn more about tourists’ actual traveling preferences, and the results can act as references for the government and concerned parties when planning and implementing policies. Furthermore, concerning the socioeconomic status of respondents, the survey only focused on gender, age, education level, occupation, income, and residence location. Future research may include more survey items, such as status of travel companions, the number of travel companions, the age of the companions, and the major aim of travel (historical heritage, natural landscape, local art and literature, etc.), to further examine if there are more aspects that affect respondents’ preferences regarding environmental resources.

## 5. Conclusions

This study performed an extended application of CEM using VR panoramic images and audio to deliver attributes evaluated and to avoid cognitive differences caused by traditional expressions via pictures and words. The preferences of local residents and tourists on travel attributes were also further examined. Perspectives on land resources (land development and utilization and the maintenance of natural landscape), cultural experiences (fish farming experience and ecological experiments), history and humanities (local landscape and architecture imagery), and industry (local product and industry promotion) were included in the evaluation model. The results demonstrated that tourists and local residents had different preferences regarding the allocation of fishing villages tourism resources. Sociodemographic and economic characteristics such as age, education, and income were also found to significantly influence tourists and local residents’ perceptions towards the environmental impacts of fishery tourism. Cognitive differences exist between tourists and local residents on various topics, including land development and utilization, local landscape and architecture imagery, as well as the incorporation of local industries and product development into villages. Less educated and older residents are inclined to increase land development and utilization by reducing natural landscapes, while tourists preferred preserving landscapes and preventing land developments. More educated residents believed that local landscape imagery was essential, but its attraction to tourists was the lowest of all the attributes. The opinions of local residents were divided on incorporating local industries and product development into villages, while tourists who were more educated, older and higher earners believed that a selling platform combining local industries and products within the villages would be attractive for tourists with similar backgrounds as them. In conclusion, to ensure the sustainable development of a tourism destination, we must incorporate stakeholders’ ideas as well as consider tourists’ needs and preferences in tourism environments together with the present situation of leisure resources, environmental characteristics, and the management of the destination.

In the future, planners of fishing village development should consider the opinions of tourists and local residents. Additionally, when there are plans to adjust the allocation of tourism environment resources, it is important to avoid a result that is merely beneficial to a minority of residents. Thus, local residents should be invited to discuss the adjustments. In such discussions, planners should explain the reasons for the reallocation of environmental resources and the benefits of the adjustments to resolve disagreements with residents and reduce any impediments in implementing policies.

## Figures and Tables

**Figure 1 ijerph-17-07306-f001:**
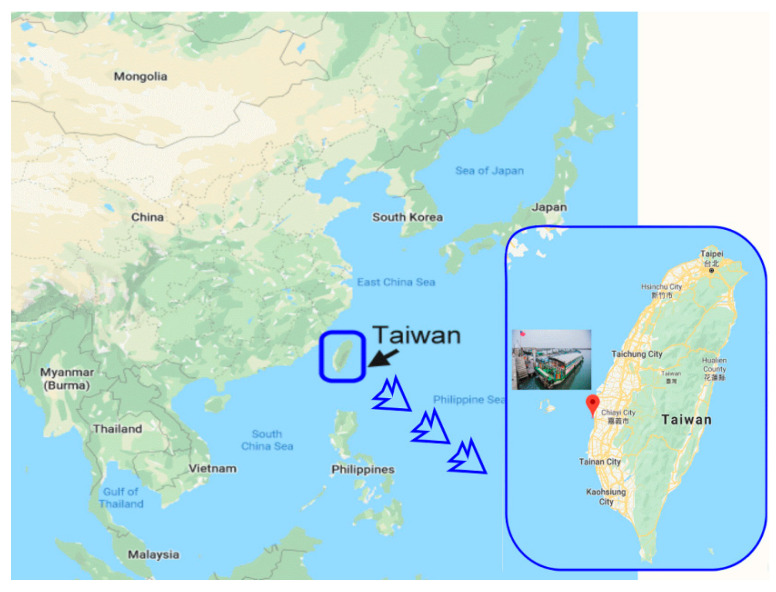
Map of the study area.

**Figure 2 ijerph-17-07306-f002:**
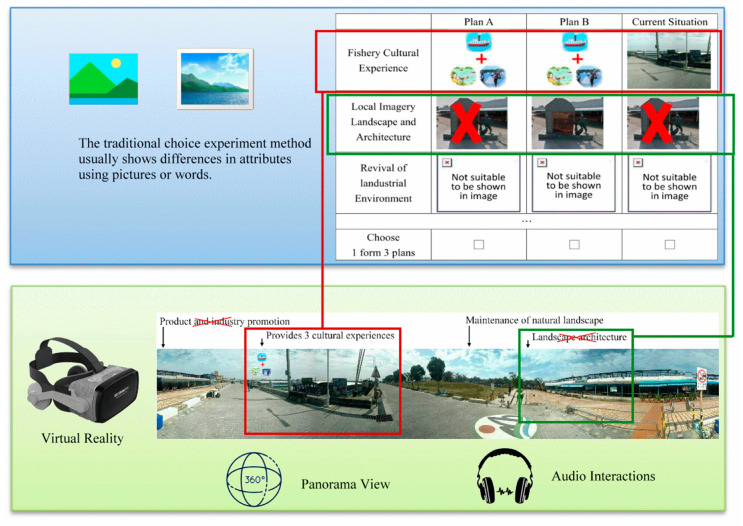
Observed differences between the traditional choice experiment method and after the use of virtual reality tools.

**Table 1 ijerph-17-07306-t001:** Attributes and attribute levels of fishing village tourist attractions.

Attributes	Levels	Variable	Number of Levels
Land use planning(LUP)	1. Maintaining the status quo	LUP±	3
	2. Increase land use and planning	LUP+	
	3. Maintenance of natural landscape	LUP−	
Cultural experience(CE)	1. Maintaining the status quo	CE	3
	2. Provides two cultural experiences	CE+	
	3. Provides three cultural experiences	CE++	
Landscape architecture(LA)	1. Maintaining the status quo	LA±	2
	2. Increase landscape architecture	LA+	
Product and industry promotion(PIP)	1. Maintaining the status quo	PIP	2
	2. Product and industry promotion	PIP+	
Evaluation attributes Visitor: Extra travel time (min) Local residents: Tourist Site Maintenance Fund	Visitor/Local residents		5
	1. Maintaining the status quo		
	2. 100 min/100 NTD per month		
	3. 150 min/150 NTD per month		
	4. 200 min/200 NTD per month		
	5. 250 min/250 NTD per month		

The superscript ± describes the attribute level included in the basic alternative. The superscript + (++) indicates an increase (strong) compared with the basic alternative and the superscript (–) indicates a reduction.

**Table 2 ijerph-17-07306-t002:** Sociodemographic and economic characteristics of the respondents.

Description		Visitors		Local Residents	
		Number	%	Number	%
Gender	Male	293	47.9	88	51.8
	Female	319	52.1	82	48.2
Marital status	Single	101	16.5	21	12.4
	Married	511	83.5	149	87.6
Education	High school	54	8.8	69	40.6
	University	392	64.1	84	49.4
	Master’s	166	27.1	17	10.0
Age (years)	20–29	158	25.8	14	8.2
	30–39	241	39.4	47	27.6
	40–49	131	21.4	47	27.6
	50–59	66	10.8	49	28.8
	≥60	16	2.6	13	7.6
Monthly income (NTD) ^a^	<25,000	127	20.8	89	52.4
	25,001–50,000	377	61.6	76	44.7
	≥50,001	108	17.6	5	2.9

^a^ NTD: new Taiwan dollar (1 NTD = 0.033 USD).

**Table 3 ijerph-17-07306-t003:** Results of the conditional logit model.

Variables and Levels	Visitors		Local Residents	
	Coeff.	t-Statistic	Coeff.	t-Statistic
ASC	2.183	6.22 ***	1.719	2.82 ***
LUP+	−0.886	−11.59 ***	0.251	1.93 *
LUP−	1.57	15.48 ***	0.417	2.67 ***
CE+	1.589	18.99 ***	1.095	7.90 ***
CE++	2.208	21.10 ***	1.383	8.05 ***
LA+	0.232	6.69 ***	0.555	8.70 ***
PIP+	0.457	12.83 ***	0.621	9.75 ***
Willingness to Pay/Leisure Attractiveness	−0.009	−13.93 ***	−0.004	−4.25 ***
Number of choice sets	3060		850	
Log-likelihood ratio	−1791.69847		−573.27	

* *p* < 0.1; *** *p* < 0.01; alternative specific constant (ASC) for the status quo.

**Table 4 ijerph-17-07306-t004:** Results of random parameter logit model.

Variables and Levels			Visitors				Local Residents			
	Coeff.	t-Statistic	Coeff. Std	t-Statistic	ETT	Coeff.	t-Statistic	Coeff. Std	t-Statistic	WTP
ASC	1.968	3.75 ***	0.996	1.74 *	-	1.28	1.08	1.11	1.98 **	-
LUP+	−1.365	−10.20 ***	1.384	9.26 ***	−109.71	0.735	2.31 **	1.996	5.65 ***	72.07
LUP−	2.291	12.78 ***	1.098	7.39 ***	184.16	1.059	2.66 ***	2.796	4.47 ***	103.81
CE+	2.259	16.04 ***	0.424	1.98 **	181.61	2.44	6.72 ***	0.657	2.10 **	239.25
CE++	3.164	16.22 ***	0.661	3.46 ***	254.32	3.104	6.07 ***	1.461	3.92 ***	304.35
LA+	0.299	5.73 ***	0.501	5.33 ***	24.06	1.133	6.07 ***	1.081	4.92 ***	111.05
PIP+	0.673	10.44 ***	0.652	5.97 ***	54.12	1.333	6.40 ***	1.187	4.58 ***	130.71
Willingness to Pay/Leisure Attractiveness	−0.012	−12.30 ***	−	−	−	−0.01	−4.21 ***	−	−	−
Number of choice sets	3060					850				
Log-likelihood ratio	−1686.84844 ***					−512.88136 ***				
Chi-square	3349.81032					841.87816				

* *p* < 0.1; ** *p* < 0.05; *** *p* < 0.01. ETT: extra travel time (minutes); WTP: willingness to pay; ASC: alternative specific constant.

**Table 5 ijerph-17-07306-t005:** Cross-analysis of tourist socioeconomic background and environment attractiveness.

Visitor		LUP+		LUP−		CE+		CE++		LA+		PIP+	
Social Characteristic	obs	Mean	*t*-Statistic/F-Test	Mean	*t*-Statistic/F-Test	Mean	*t*-Statistic/F-Test	Mean	*t*-Statistic/F-Test	Mean	*t*-Statistic/F-Test	Mean	t-Statistic/F-Test
Single	101	−119.81	−1.573	178.66	−0.742	180.38	−1.187	238.07	−9.088 ***	26.86	1.547	52.28	−0.198
Married	511	−106.52		182.89		181.98		256.63		23.82		52.85	
High school	54	−85.83	12.44 ***	163.07	6.027 ***	181.36	5.153 ***	254.59	3.196 **	23.78	0.123	52.37	8.741 ***
University	392	−103.48		182.49		182.64		254.72		24.15		49.79	
Master’s	166	−128.52		187.71		179.65		250.51		24.89		59.88	
20–29 years old	158	−127.83	10.957 ***	186.36	2.91 **	180.66	1.443	255.15	3.704 ***	21.65	3.599 ***	54.63	5.628 ***
30–39 years old	241	−110.17		185.66		182.89		251.8		23.52		48.12	
40–49 years old	131	−104.3		179.39		181.28		252.7		25.65		52.3	
50–59 years old	66	−82.77		170.86		180.76		259.87		28.24		63.54	
More than 60 years old	16	−40.96		158.54		181.89		245.65		35.78		63.26	
Less than 25,000 NTD	127	−103.37	0.574	180.88	0.09	183.43	3.613 **	252.52	0.539	21.54	1.909	42.53	14.079 ***
25,001–50,000 NTD	377	−110.61		182.29		181.67		253.51		24.98		54.34	
More than 50,001 NTD	108	−108.37		183.4		179.88		254.99		25.3		59.25	
North	157	−102.2	1.591	179.22	0.447	180.78	18.954 ***	267.69	16.358 ***	22.04	1.327	53.06	1.663
East	76	−122.08		183.24		175.27		259.44		23.88		56.32	
West	266	−109.41		184.25		184.48		249.6		25.07		53.49	
South	113	−107.12		180.79		180.85		239.33		26.01		48.21	

** *p* < 0.05; *** *p* < 0.01.

**Table 6 ijerph-17-07306-t006:** Cross-analysis of resident socioeconomic background and willingness to pay for environment attributes.

Local Residents		LUP+		LUP−		CE+		CE++		LA+		PIP+	
Social Characteristic	obs	Mean	*t*-Statistic/F-Test	Mean	*t*-Statistic/F-Test	Mean	*t*-Statistic/F-Test	Mean	*t*-Statistic/F-Test	Mean	*t*-Statistic/F-Test	Mean	*t*-Statistic/F-Test
Single	21	−40.17	−3.124 ***	98	−0.094	235.13	−1.046	318.24	1.156	96.46	−1.011	112.49	−1.453
Married	149	62.57		102.83		244.7		300.1		112.69		138.81	
High school	69	128.58	28.938 ***	62.44	6.598 ***	248.11	1.83	311.02	1.43	97.45	2.699 *	125.82	1.057
University	84	15.69		109.91		240.9		299.29		116.64		140.29	
Master’s	17	−100.63		225.8		237.82		282.12		135.01		151.7	
20–29 years old	14	−44.78	12.559 ***	262.57	6.986 ***	248.82	0.885	307.53	0.576	117.84	0.218	111.72	3.685 ***
30–39 years old	47	−29.72		157.79		240.5		312.33		108.44		159.97	
40–49 years old	47	51.84		68.91		241.49		291.81		116.99		119.77	
50–59 years old	49	133.9		54.83		244		300.51		106.34		146.89	
More than 60 years old	13	115.79		27.84		254.25		305.57		104.72		87.3	
Less than 25,000 NTD	89	68.41	1.868	89.43	0.587	242.61	1.284	305.39	0.442	107.76	0.45	144.1	1.198
25,001–50,000 NTD	76	32.81		114.28		245.68		300.35		112.43		125.32	
More than 50,001 NTD	5	−20.44		147.01		226.82		278.22		136.42		139.17	

* *p* < 0.1; *** *p* < 0.01.

**Table 7 ijerph-17-07306-t007:** Empirical projection results on potential category model.

Attributes and Levels Parameters	Category 1 (73.10%)			Category 2 (26.90%)		
	Coefficient	t-Value	WTP	Coefficient	t-Value	WTP
Constant	−25.1	0	-	−12.54	−4.23	-
LUP+	−0.34	−3.38 ***	−52.19	−7.89	−4.79 ***	−145.93
LUP-	1.49	10.76 ***	228.84	3.39	6.59 ***	62.69
CE+	2.17	16.51 ***	333.62	0.22	0.82	-
CE++	2.73	17.29 ***	420.3	0.91	2.36 **	16.85
LA+	0.21	4.44 ***	31.6	1.25	4.32 ***	23.18
PIP+	0.63	12.11 ***	97.55	1.26	4.34 ***	23.36
FUND	−0.01	−7.9 ***	-	−0.05	−5.31 ***	-
Constant	−0.42	−1.06				
Rich natural landscape	0.69	3.24 ***				
More entertainment facilities	2.38	8.28 ***				
More cultural experience	1.83	8.61 ***				
Increase in local architecture imagery	−0.43	−1.8 *				
Incorporation of local industry and product promotion	0.6	2.82 ***				
Married	0.49	2.16 **				
AGE ≤ 49	−0.66	−1.96 *				
N of choice sets	3060					
Log-likelihood ratio	−1562.247					
Chi squared (degree of freedom)	3599.012 [24]					

* *p* < 0.1; ** *p* < 0.05; *** *p* < 0.01.

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
