# Peer review of "The Construction and Validation of a Sustainable Tourism Development Evaluation Model"

_ijerph, 2020, doi:10.3390/ijerph17197306_

Round 1

Reviewer 1 Report

See attached file

Author Response

Thank you very much for your insightful comments and suggestions. We believe your comments and suggestions are appropriate and useful to us in order to improve considerably the quality of the manuscript. We have revised our paper in light of your comments and instructions. Please refer to attached document.

Reviewer 2 Report

The paper is well written, easy to read and timely assessment of sustainable tourism. Here are my comments: 

  1. Page 3, Line 99 - needs citation
  2. Page 3, Line 133 - needs citation
  3. Page 8, Figure 2 - text missing - "The traditional choice experiment method.."
  4. Page 9, Table 3 - ASC variable needs to be defined
  5. Page 10, Table 3 - FUND variable needs to be defined

Author Response

(The authors gave the same response as above.)

Reviewer 3 Report

Congratulations! I really enjoyed your article. The paper follows the best Academia rules.

As you refer in point 5.3. (Limitations and future research) I hope that you are able to include, in your next article, more survey items such as the number of travel companions, the age of companions and the major aim of the travel. In fact, being the same person, a tourist has different travel behaviours, depending on with whom is travelling; it is not the same to travel with friends or only with the consort  or to travel with children who become often annoyed with grown-up people activities.

Author Response

(The authors gave the same response as above.)
